# Microbial Contamination of Photographic and Cinematographic Materials in Archival Funds in the Czech Republic

**DOI:** 10.3390/microorganisms10010155

**Published:** 2022-01-12

**Authors:** Sabina Purkrtova, Dana Savicka, Jana Kadava, Hana Sykorova, Nikola Kovacova, Dominika Kalisova, Tereza Nesporova, Martina Novakova, Barbora Masek Benetkova, Lucie Koukalova, Stepanka Boryskova, Blanka Hnulikova, Michal Durovic, Katerina Demnerova

**Affiliations:** 1Department of Biochemistry and Microbiology, Faculty of Food and Biochemical Technology, University of Chemistry and Technology Prague, 5 Technická, 166 28 Prague, Czech Republic; Dana.Savicka@vscht.cz (D.S.); Jana.Kadava@vscht.cz (J.K.); Hana.Sykorova@vscht.cz (H.S.); Kovacova.Nikola@seznam.cz (N.K.); kalisovd@vscht.cz (D.K.); Tereza.Nesporova@vscht.cz (T.N.); 2Department of Chemical Technology of Monument Conservation, Faculty of Chemical Technology, University of Chemistry and Technology Prague, 5 Technická, 166 28 Prague, Czech Republic; Martina.Novakova@vscht.cz (M.N.); Barbora.Masek.Benetkova@vscht.cz (B.M.B.); Lucie.Koukalova@nacr.cz (L.K.); 3Preservation Department, National Archives Czech Republic, 2257/4 Archivní, 149 00 Prague, Czech Republic; Stepanka.Boryskova@nacr.cz (S.B.); Blanka.Hnulikova@nacr.cz (B.H.)

**Keywords:** photographic materials, cinematographic materials, archival funds, Czech Republic, microbial contamination, fungal contamination, bacterial contamination, cultivation methods, MALDI-TOF MS

## Abstract

In this study we investigated the microbial contamination of 126 samples of photographic and cinematographic materials from 10 archival funds in the Czech Republic. Microorganisms were isolated from the light-sensitive layer by swabbing it with a polyurethane sponge. Microbial isolates were identified by MALDI-TOF MS (bacteria) or by phenotype testing and microscopy (fungi). Bacterial contamination was more abundant and more diverse than fungal contamination, and both were significantly associated with archives. The most frequently isolated fungal genera were *Cladosporium*, *Eurotium*, *Penicillium, Aspergillus* and *Alternaria.* The most frequently isolated bacteria were Gram-positive genera such as *Staphylococcus*, *Micrococcus*, *Kocuria*, *Streptococcus* and *Bacillus*. This bacterial and fungal diversity suggests that air is the main vehicle of contamination. We also analysed the impact of the type of material used for the carrier (paper, baryta paper, cellulose acetate and nitrate or glass) or the light-sensitive layer (albumen, gelatine, collodion and other) on the level and diversity of microbial contamination. Carriers such as polyester and cellulose nitrate may have a negative impact on bacterial contamination, while paper and baryta paper may have a partially positive impact on both fungal and bacterial contamination.

## 1. Introduction

Photographic and cinematographic materials have become an integral part of our lives over the past 200 years. They are art forms and a testimony to the past of human society and therefore need to be preserved for future generations.

Photographic and cinematographic materials are very diverse in their composition. These materials are—with a few exceptions such as, e.g., digital photography—composed of two parts, one of which is the carrier, on which a light-sensitive layer is applied in one or more layers [1]. The light-sensitive layer is traditionally defined as an emulsion of a light-sensitive substance (optical information repository) and a binder [2].

By illuminating the light-sensitive layer, a negative image is produced, which is tonally and colour-inverse to the positive image. The development (even multiple) of a positive image (the image as perceived by the human eye) occurs by lighting photographic paper or photographic film through a negative image. The individual photographic and cinematographic techniques used throughout history differ not only in the combination of materials used for the carrier and light-sensitive layer of both the negative and positive images, but also in the methods used for the negative and positive processes themselves [3,4].

The light-sensitive substance is typically composed of microcrystals of silver halides (AgCl, AgBr and AgI), which have rapidly replaced other substances (e.g., bituminous dust in lavender oil, heliography method; platinum with iron salts, platinotype method; potassium ferricyanide and ferric ammonium citrate, cyanotype method). Silver cations are reduced to neutral silver atoms (latent image spots) by the absorption of electromagnetic radiation, which are reduced to metallic silver by means of a reducing agent (developer) [5]. 

Silver halides absorb light only in the range of short wavelengths (ultraviolet and blue light) and thus form only a black-and-white image. The absorption of other wavelengths (including those for the three primary colours blue, red, and green) is made possible by the addition of special sensitizers (dyes) to the photographic emulsion [6]. For colour photography, orthochromatic emulsion, coloured potato starch grains covered with panchromatic emulsion (autochrome plates) or stratification of three emulsion layers sensitive to basic colours (modern colour photomaterials) were over time gradually used as a light-sensitive layer [1,6].

The light-sensitive substance is resuspended in a suitable binder material such as collodion (a solution of nitrocellulose in diethyl ether and ethanol), albumen or gelatine (proteins) with admixtures of other substances (sensitizers, antifogants and hardeners) [7]. The most commonly used binder is gelatine, a protein obtained by extraction from animal skin, bones and connective tissue [2], which has not yet been replaced by synthetic polymers [8].

The carrier must have suitable physico-chemical properties such as strength or flexibility, stability, low coefficient of friction and good adhesion, and no effect on the photochemical properties of the light-sensitive layer (photochemical inertness) throughout the photographic process or its use [3]. Over time, photographic carriers that were made of metal, glass, uncoated or coated photographic paper (coated by resin, polyethylene, barium sulphate or lacquered) or synthetic polymers (cellulose nitrate or acetate, polyethylene terephthalate, polyester) were used [7]. Synthetic polymers, which are flexible, enabled further development—especially in cinematography (e.g., celluloid film, Kodachrome) [4,7].

Photographic and cinematographic materials, composed of both inorganic and organic substances, may be subject to both physico-chemical and microorganism degradation, for which organic substances are a suitable source of necessary nutrients and energy.

Physico-chemical degradation processes include, for example, the spontaneous formation of neutral silver atoms in unexposed microcrystals of silver halides (fog creation); the formation of volatile substances in photographic papers during prolonged storage; or the destruction of chemically unstable and highly flammable cellulose nitrate or spontaneous deacetylation of cellulose triacetate to acetic acid (so-called vinegar syndrome due to the typical odour of damaged films) [1,2,3]. Vinegar syndrome causes colour fading, deformation and hardening of the material and leads to its partial or even complete loss [9].

Microorganisms produce many enzymes that degrade the organic polymers present in these materials, as well as other chemicals that can degrade materials, e.g., lipases and esterases degrade some polyesters, proteases degrade proteins such as albumen, amylases degrade starch, cellulases degrade cellulose derivatives and gelatinases liquefy gelatine [7]. Microorganisms are ubiquitous and their main source is primarily the air or human skin when in contact with photographs. Not only eukaryotic microorganisms such as micromycetes (fungi), but also prokaryotic microorganisms such as bacteria (and archaebacteria) are able to contaminate and colonize the surfaces of photographic and cinematographic materials.

Photographic and cinematographic materials are therefore very sensitive to various factors, especially temperature, humidity, long-term exposure to light and UV radiation, as well as the presence of chemicals and biological contaminants or poor treatment by insufficiently trained staff [1,10]. Therefore, storage conditions and handling methods play a major role in their protection [7]. These conditions must be chosen so that they are not suitable for the growth of microorganisms or the physico-chemical degradation of materials, but at the same time do not damage these materials and are comfortable for professionally trained personnel to work comfortably in. At the same time, further microbial contamination must be prevented (e.g., by air filtration) and the original contamination minimized.

The basis for long-term storage is an enclosed space in which constant conditions can be maintained. Necessary are cleanliness, good ventilation and, ideally, storage in the dark [11]. The recommended storage conditions differ for single materials [12], but generally lower temperatures (while controlling relative humidity) are more suitable to prevent physico-chemical processes, e.g., for storing photographic materials, the maximum suitable temperature is 21 °C and the optimal relative humidity is 30–50%, where higher values increase the activity of microorganisms and lower values cause the drying and embrittlement of photographs [11,12]. Photographic and cinematographic materials should also be stored in a sufficiently spacious protective packaging made of a suitable material—typically paper, plastic (but not for nitrate and acetate films due to insufficient air circulation) or metal (metal boxes for cinematographic films) [11,13].

The basic principle of working with photographic and cinematographic material is a clean and dry working environment and careful handling of the material [11]. Hands must be properly washed and dried, as dirt, sweat and oil can leave permanent imprints, or ideally white cotton or latex gloves for work would be used. It is important to avoid touching the image of a photograph or film footage directly [11,13]. Both mechanical cleaning methods (dry mechanical cleaning, 70% ethanol cleaning) and chemical disinfection methods (using chemical disinfectants as alcohols, phenols, essential oils, alkylating agents or quaternary ammonium compounds, but not water or aqueous solutions) can be used to remove already present microbial contamination [7]. The chosen method must be sufficiently effective against contaminating microorganisms, but at the same time sensitive to the treated material, on which it must not have a harmful effect, and also safe for humans and the environment [14]. Mechanical cleaning may not be sufficient for eliminating microbial infestation of these materials, unlike chemical disinfection [14].

An important aspect of the protection of photographic and cinematographic materials are then microbial inspections directly in the archives, with both direct inspections of material surfaces and microbial air analysis (free sedimentation, aeroscope). These microbial controls provide information on the degree of risk of biodegradation of deposited materials [15]. 

While the importance of physical and chemical factors in the archiving of photographic and cinematographic materials has been intensively studied, microbial contamination of these materials has not yet been sufficiently described, and there are very few studies in this field.

However, knowledge of microorganisms that predominantly contaminate photographic and cinematographic materials makes it possible to find and evaluate the optimal methods for protecting these materials from microbial contamination and degradation.

Two different approaches can be applied for the investigation of microbial diversity: isolating the microorganisms present as a living microbial culture using cultivation methods or isolating and analysing the microbial DNA or RNA present using culture-independent methods (a metagenomic approach). In cultivation methods, the cultivation conditions determine the taxonomic diversity and number of isolated microbial colonies [16]. On the other hand, only viable microbial cells can be cultivated under given cultivation conditions. These viable microbial isolates can be further studied for their true ability to degrade archival materials and their sensitivity to decontamination procedures. In the metagenomic approach, sequencing of 16S rRNA genes for bacteria and the ITS region for fungi allows the identification of not only viable—but non-cultivable—microorganisms, but also dead microorganisms (e.g., the DNA of microorganisms degraded by decontamination processes) [17]. From methods reducing this negative aspect, the use of isolated RNA as a marker of microorganism viability in some studies on microbial diversity, or on monitoring the effect of conservation treatments, has been recently proposed [18,19,20]. However, to determine the microorganism’s ability to degrade materials then requires a further targeted DNA-based study of genes involved in the appropriate metabolic pathway [21], or to detect pertinent metabolites on photographs [22]. The sampling for both these approaches can be destructive (as material rinsing or homogenization or its directly placing on agar media) [9,16,22,23] or non-destructive (such as swabbing or membrane pressing) [16,20].

While these new metagenomic approaches are innovative and provide a broad new insight into the microbiome of different culture heritage specimens, their performance also requires very specific equipment and methods, conducted by highly trained experts, compared to the easily conducted cultivation methods. The identification of microorganisms isolated using cultivation methods can be also performed by two different approaches. The first approach involves phenotype-based methods, such as classical methods analysing macroscopical and microscopical features and biochemical testing, or rapid methods such as protein spectrum analysis by MALDI-TOF MS [23,24]. The second approach covers genotype-based methods such as the sequencing of 16S rRNA for bacteria [25] or the ITS region for fungi [26]. While the classical phenotype methods are laborious, material- and time-consuming, and the genotype-based methods require multistep analysis (DNA isolation, PCR amplification and PCR amplicon sequencing), MALDI-TOF MS is a rapid and robust method, providing results in a real time [27]. This method has, furthermore, been proven previously—with some limitations—to be applicable for the identification of the microbiome of contaminated historical books, especially for their bacteriome [20].

Studies of microbiome diversity in contaminated photographic or cinematographic materials are also scarce for the cultivation-based approach and, if performed, they cover only a limited number of samples (up to 16 samples [28]). These studies include, for example, research on microbial contamination in cinematographic collections in Spain [28], Portugal [29], Italy [30,31] or Cuba [9], or research on albumen photographs in Slovakia [16].

The aim of this work was, therefore, to use cultivation methods to determine the degree and diversity of microbial contamination of contaminated photographic and cinematographic materials from ten selected archives in the Czech Republic, and to define the impact of sample features (type of material, type of light-sensitive layer and carrier) on this diversity. Another aim was to introduce the MALDI-TOF MS phenotypic method as a relatively fast and reliable method for the routine identification of contaminating bacterial species.

## 2. Materials and Methods

### 2.1. Tested Archives and Samples

In this study, we investigated samples of contaminated photographic and cinematographic materials from a total of ten different archive stations in six entities of archival funds in the Czech Republic, including two State Regional Archives (SRA), three State District Archives (SDA) and one National film archive (NFA) in 2019–2021. The investigated archival funds (archives) were (the used abbreviations given in the brackets): SDA Beroun (March 2019, “A”), SDA Mlada Boleslav (April 2019, “B”), NFA Hradistko (August 2019, “C”), SRA Litomerice—Litomerice (“D1”) and Decin (“D2”) stations (both December 2019), SRA Praha (July 2020, “E”), SDA Plzen—Nepomuk station (both September 2020, “F”) and SRA Trebon—Jindrichuv Hradec (“G1”), Ceske Budejovice (“G2”) and Trebon (“G3”) stations (all July 2021). Across ten archive stations, a total of forty-two depositary rooms were investigated (maximum of 9 depositaries from any one archive station), from which a total of 126 samples of photographic (90 samples of photographic positives, 19 samples of photographic negatives) and cinematographic materials (17 samples) from 80 different archival items (as photoalba) were chosen for analysis. To investigate the microbiome present, samples suspected of microbial contamination with different properties (photographic positives or negatives, cinematographic negatives; different materials of light-sensitive layer and carrier) were chosen. The possible presence of microbial contamination on the light-sensitive layer was evaluated by a digital microscope Keyence VHX 6000 S (Keyence, JP), as most of samples were not visibly deteriorated.

### 2.2. Isolation of Microorganisms from the Light-Sensitive Layer

The light-sensitive layer in all the samples was analysed by a specific swabbing method, previously developed [32] to be the most efficient and sensitive method for the tested material. The light-sensitive layer was swabbed by gently rolling a dry polyurethane sponge (PUR-Blue Swab Sampler with Dry Large Tip Swab, World Bioproducts, Libertyville, IL, USA) across its surface. After transport to the laboratory, the sponge was lightly moistened with sterile saline (0.90% NaCl) and gently rolled over the surface of Petri plates (average 90 mm) with solidified culture media (Malt Extract Agar—MEA, Dichloran-Glycerol Agar—DG18, Plate Count Agar—PCA) in a crosswise manner. The primary cultivation was performed for yeasts and fungi on MEA (Malt Extract Agar, Oxoid, Hampshire, UK) and for xerophilic fungi and osmophilic yeasts (from material with the water activity 0.95 or lower) on DG18 (Dichloran-Glycerol Agar Base, Oxoid, Hampshire, UK; glycerol, Penta, Prague, Czech Republic) at 22 °C under light for 30 days. The growth was subsequently controlled for 7, 14 and, finally, 30 days. The primary cultivation for bacteria was performed on PCA (Plate Count Agar, Merck, Darmstadt, Germany) at 30 °C for 3–5 days. 

After this primary cultivation, the colony-forming units (CFU) of fungi grown on MEA and DG18, and the CFU of bacteria grown on PCA were analysed according to their phenotype profiles. The CFU of single phenotype profiles were counted, numbered and sub-isolated. For bacteria, at least one third of the CFU of a single phenotype profile was sub-isolated for further identification. 

### 2.3. Phenotype Identification of Fungi

Fungal isolates, isolated on MEA and DG18, were sub-isolated on MEA, if necessary, and identified by their macromorphological features and structures and micromorphological structures [33,34,35,36,37]. Macromorphological features included the evaluation of colony size, structure and colour of its surface and reverse, its growth rate and ability to grow on media with limited water activity (DG18) or at higher temperatures, the presence of exudate, and its appearance under a magnifying glass—such as the presence of certain structures such as, e.g., sclerotia, synnemata, pustules, pycnids and pigment formation. For micromorphological features, they were evaluated according to the type of conidiogenesis, the size, shape and surface of conidia, the surface of conidiophores or sporangiophores, the presence of chlamydospores, the method of branching hyphae and others. Most of the isolated fungi were classified on the genus level, while the species identification was determined only for some isolates.

### 2.4. MALDI-TOF MS Identification of Bacteria

Bacterial isolates, isolated on PCA, were sub-isolated on CSB agar (Columbia agar with 5% sheep blood, RTU agar plates, Bio-Rad, Hercules, CA, USA) and cultivated at 30 °C for at least 24 h to achieve sufficient growth, and then identified by MALDI-TOF MS (matrix-assisted laser desorption ionization–time of flight mass spectrometry) method. 

MALDI-TOF MS identification of bacterial isolates was performed with an Autoflex Speed mass spectrometer (Bruker Daltonics, Bremen, Germany) and using the MALDI Biotyper 3.1 database (Bruker Daltonics, Bremen, Germany). The samples were spotted on a polished steel plate MTP 384 (Bruker Daltonics, Bremen, Germany) in at least three spots (technical replicates). The matrix was a solution of alpha-cyano-4-hydroxycinnamic (HCCA) acid (Bruker Daltonics, Bremen, Germany) at a concentration of 10 mg/mL^−1^ in organic solvent (dissolved by vortexing or shaking at room temperature). One ml organic solvent was prepared from 500 µL acetonitrile, 250 µL 10% trifluoroacetic acid and 250 µL nuclease-free water. All chemicals were purchased from Sigma-Aldrich (St. Louis, MO, USA) except the nuclease-free water (Thermo Fisher Scientific, Waltham, MA, USA).

Three methods for sample preparation were used, which differ in the efficiency of their extraction of intracellular proteins. The required limit for species identification was a score value of more than 2.0 and species consistency (an evaluation of (++) (A) or (+++) (A)). The required limit for the group species identification was a score value more than 2.0 and genus consistency (an evaluation of (++) (B) or (+++) (B)). The first-choice method was eDT, as it is efficient for a broader spectrum of bacterial species and consumes less time than the extraction method. The second-choice methods were DT or the extraction method EX EtOH/FA (preferably for Gram-positive bacteria).

In the direct transfer (DT) method, a small amount of freshly grown culture was applied to a spot on a steel plate and allowed to dry (no longer than 30 min generally or 60 min for Gram-positive bacteria). In the extended direct transfer (eDT) method, this dried culture was then covered with 2 µL of 70% solution of formic acid and allowed to dry for at least 15 min. In the extraction (Ex EtOh/FA) method, the liquid extract of the intracellular proteins was spotted. To prepare this extract, freshly grown cultures were resuspended in 300 µL of sterile distilled water, mixed with 900 µL of absolute ethanol and centrifuged (2 min, 13,000 rpm). The pellet (dried at room temperature for at least 15 min) was resuspended in 50 µL of 70% formic acid and then in 50 µL of acetonitrile. This mixture was centrifuged (2 min, 13,000 rpm) and 2 µL of supernatant were applied onto a spot and, after drying, immediately covered by matrix. For the spectrometer calibration, Bruker Bacterial Test Standard (Bruker BTS, Bruker Daltonics, Bremen, Germany) was applied onto two spots (0.5 and 1 µL) and, after drying, immediately covered by matrix. All sampled spots, including for Bruker BTS, were covered with HCCA matrix (up to 2 µL) and allowed to dry for at least 15 min at room temperature, until the spotted mixture colour turned yellow.

### 2.5. Statistical Analysis

The statistical analyses were performed in R language (version 4.1.1) [38] using the package ‘stats’ for Pearson’s Chi-squared test and ‘corrplot’ for the visualisation [39].

## 3. Results

### 3.1. Distribution of Sample Types and Materials

In this study we investigated 126 different samples of suspectedly contaminated photographic and cinematographic materials taken from 80 archival items deposited in 10 different archive stations from a total of 42 depositaries. As single samples taken from one archival item were predominantly of a different composition, they were analysed independently. 

The samples differed in their type (photographic positives or negatives, cinematographic negatives), and in the material of the light-sensitive layer and the carrier. The characterisation of the analysed samples from single archives is summarised in Appendix A. Generally, most samples were photographic positives (PHPOS; 71%, 90 samples), while photographic negatives (PHNEG; 15%, 19 samples) and cinematographic materials (CIN; 13%, 17 samples) were less frequent. All photographic negatives and cinematographic negatives and 48% of photographic positives had a gelatine (GEL) light-sensitive layer (63% of all samples). Other light-sensitive materials were present only in photographic positives, such as albumen (ALB; 23% in PHPOS; 17% of all), collodion (COL; 19%; 13% of all) or others (OTH; 10%; 7% of all—including autotype, ozotype, collotype, cyanotype, platinotype, salt paper and gumoil printing). The carrier materials differed for CIN and PHNEG in comparison to PHPOS. For CIN, it was dominantly cellulose acetate (88%; CA) or less frequently polyester (12%; PES); for PHNEG it was glass (48%; GL), cellulose acetate (21%; CA) and cellulose nitrate (26%; CN), or very rarely polyester (5%; PES). For PHPOS, baryta paper (51%; BP) and paper were used (49%; PAP). However, for ALB and OTH light-sensitive layers, PAP was the only carrier. In contrast, BP was dominant as the carrier for GEL (79%) or COL (71%). 

In six archives we were able to analyse all three kinds of samples (PHPOS, PHNEG, CIN). In one archive, only samples of PHNEG and PHPOS were analysed, and in three archives, only samples of PHPOS.

### 3.2. Fungal Contamination

From 126 analysed samples, 60% were positively tested for fungal contamination (≥1 CFU isolated). We isolated 120 isolates, from which 83% (99 isolates) were identified at least on the genus level (12 genera) and 17% (21 isolates) were not identified, as they did not sporulate. The relative frequencies of samples positive for fungal contamination and fungal genera present from single archives are summarised in Table 1.

The most frequent fungal genera isolated from the archives were *Cladosporium* sp. (90% of archives), *Eurotium* sp. and *Penicillium* sp. (both 80%), *Aspergillus* sp. (50%) and *Alternaria* sp. (30%). Other fungal genera or species were rare, detected only in one of the tested archives (10%), such as *Arthrinium* sp., *Epicoccum nigrum*, *Geomyces* sp. and *Tritirachium oryzae* (all in archives C), *Aureobasidium pullulans* (in G3), or *Phoma* sp. and *Wallemia sebi* (in D1). Archives differed not only in the frequency of positive samples from 100% (archives G2) to 28% (archives B), but also in the diversity of isolated fungal contamination (up to 9 fungal genera isolated in archives C).

The fungal load for single samples was not very high. While from 40% of all samples no fungal CFU were isolated, from 52% of samples only 1–5 CFU (1 CFU—25%, 2 CFU—14%, 3–5 CFU—14%) were isolated. More than 5 CFU were isolated from 8% of samples (6–10 CFU—6%, 11–36 CFU—3%). Four outlier samples loaded with more than 10 CFU were spread across three different archives: PHPOS (G3, GEL/BP—12 CFU; D1, GEL/PAP—13 CFU; E, OTH (cyanotype)/PAP—18 CFU) or PHNEG (E, GEL/PES—36 CFU).

To analyse the effects of single factors (archives, type of material, the material of the light-sensitive layer and the carrier) on the total fungal load, the samples were categorised by these factors and correlated with categories of total fungal load. The total fungal load (fungal CFU isolated from one sample) was categorised in four levels: 0 CFU as no contamination; 1–5 CFU as low contamination; 6–10 CFU as moderate contamination; and 11–36 CFU as high contamination. 

A maximum of three genera were isolated in one sample, except the two most loaded samples (18 CFU—5 genera, 36 CFU—4 genera). There was a statistically significant association (Pearson’s Chi-squared test, *p* = 0.05) between the total fungal load and the number of fungal genera present (*p* < 0.0001). Moderate fungal contamination (6–10 CFU) was positively associated with the presence of two or three fungal genera, while a high level (11–36 CFU) was associated with the presence of three, four or five genera.

Regarding the relative frequencies of total fungal load categories in different groups of samples (see Figure 1), CIN was more likely to be contaminated (65% of samples) in comparison to PHNEG and PHPOS (60% and 58%, respectively). However, if contaminated, only PHPOS and PHNEG were contaminated to a moderate or high level (10% and 5%, respectively, of samples). 

No statistically significant association (Pearson’s Chi-squared test, *p* = 0.05) was found between the level of total fungal load of samples and different kinds of materials (*p* = 0.87), material of the light-sensitive material (*p* = 0.93) or carrier (*p* = 0.32). To the contrary, a statistically significant association (*p* = 0.05) was determined between the category of total fungal load and the archives from which samples were taken (*p* = 0.004).

For fungal genera, the effects of these factors were only analysed for those which were isolated in more than one sample (equal to one unique isolate), such as the genera *Cladosporium* (40 samples), *Penicillium* (18 samples), *Eurotium* (14 samples), *Aspergillus* (13 samples) and *Alternaria* (6 samples).

No statistically significant association (Pearson’s Chi-squared test, *p* = 0.05) was found between the most frequent genera present and the kinds of materials (*p* = 0.53), material of the light-sensitive layer (*p* = 0.54) or carrier (*p* = 0.77), the combination of all these (*p* = 0.71), or the number of CFU isolated from one genus in a sample (*p* = 0.60). However, as for the total fungal load, there was a statistically significant association between isolated fungal genera and archives from which samples were taken (*p* = 0.0004).

However, the standardized Pearson residuals (see Figure 2) and the corresponding relative frequencies (see Appendix A) showed some strong positive and negative associations.

The most often isolated genus *Cladosporium* (90% archives) was ubiquitous, being isolated from all types of materials of the light-sensitive layer and carrier and their combinations, except for GEL/CA (PHNEG). It contaminated all three kinds of material (CIN, PHPOS, PHNEG) similarly. The genus *Penicillium* (60% archives) was also ubiquitous, as was *Cladosporium*, being isolated from almost all material combinations except for GEL/PES (CIN) and GEL/CN (PHNEG)—but found in decreased frequencies in all group samples. 

Contrarily, the genus *Eurotium* (60% archives) was not isolated from collodion light-sensitive layers (COL/BP, COL/PAP, PHPOS) and its isolation was strong positively associated with gelatine light-layers (except for the combination GEL/PAP, PHPOS). As the genus *Eurotium* was isolated from archives A, B, C, D1, E and G3—all of which (except G3) contained samples of COL/BP or COL/PAP—this fact deserves further investigation. 

The genus *Aspergillus* (50% archives) was isolated from all light-sensitive layer materials, but not from all carriers. Its isolation was positively associated with paper carriers, such as COL/PAP and GEL/PAP (PHPOS). Due to this fact, it was not isolated from PHNEG (except for one extremely contaminated sample GEL/PES), despite the fact that these samples were present in the archives from which it was isolated. The genus *Alternaria* (30% archives) showed the same pattern, being isolated from all light-sensitive layer materials—mainly from these on PAP or BP carriers, with the highest association to ALB/PAP and OTH/PAP (PHPOS) and COL/BP (PHPOS), and not isolated from PHNEG.

Although only three samples of polyester carriers from three different archives were analysed (GEL, CIN from A and C; PHNEG from E), all these three samples were highly contaminated with fungi—all with the genus *Cladosporium*; one also with *Eurotium*; and one with *Eurotium*, *Penicillium* and *Aspergillus.*


Fungal genera were mostly isolated with 1–5 CFU in samples (low level of contamination; see Figure 2 and Appendix A). The other rare fungal genera were always isolated with one CFU (from one sample) from GEL on BP (*Geomyces* sp., *Tritirachium oryzae*, *Aureobasidium pullulans*, all PHPOS), PAP (*Phoma* sp., *Wallemia sebi*, both PHPOS), CN (*Arthrinium* sp., PHNEG) or CA (*Epicoccum nigrum*, PHNEG).

### 3.3. Bacterial Contamination

From 126 analysed samples, 75% tested positive for bacterial contamination (≥1 CFU isolated). We finally isolated 207 unique species isolates from single plates (from 350 examined isolates), of which 91% (189 isolates) were identified on at least the species group level, and 9% (18 isolates) that were not possible to identify reliably by MALDI-TOF MS. Using MALDI-TOF MS, we identified 25 different bacterial genera and 56 bacterial species. The relative frequencies of samples positive for bacterial contamination and bacterial genera present from single archives are summarised in Table 2. 

Only four genera of Gram-negative bacteria were isolated, as Gram-negative bacteria only accounted for 4% of all bacterial isolates (genus *Neisseria*—2%, genera *Acinetobacter*, *Brevundimonas*, *Sphingomonas*—all at 0.5%). 

The predominantly isolated Gram-positive bacteria formed 87% of all bacterial isolates. The most numerous group was Gram-positive cocci (58% of all bacterial isolates), dominantly connected to the normal human bacteriome of skin and mucous membranes (such as the oral cavity, pharynx, etc.) and the air bacteriome. The Gram-positive catalase positive cocci group (40%) comprised the genus *Staphylococcus* (22%, eight species isolated, normal skin bacteriome), the species *Micrococcus luteus* (9%, also recognised as an important air contaminant) and other genera closely related to it (*Kocuria*—5%, *Rothia*—2%, *Arthrobacter*—1.5%, *Kytococcus*—0.5%). The Gram-positive catalase negative cocci group (18%) was mainly composed of the genus *Streptococcus* (17.5%, six species, connected to respiratory tract), or rarely by *Leuconostoc lactis* (0.5%, ubiquitous, not a normal part of the human bacteriome).

Another very frequent group was Gram-positive rods that formed endospores (23%). Forming endospores increases their resistance to harsh environmental conditions, and they are ubiquitous—mainly isolated from soil and air, but some of them also from human skin. The genus *Bacillus* (18%, 12 species) was dominant in this group, while other genera were rather less abundant (*Paenibacillus*—3%, *Brevibacillus*, *Psychrobacillus*, *Sporosarcina*—all 0.5%). Gram-positive filamentous bacteria (5%) of the orders *Actinomycetales* (the genera *Dermacoccus*—3%, and *Actinomyces*, *Brevibacterium*, *Oerskovia*, *Gordonia*, all 0.5%) and *Streptomycetales* (the genus *Streptomyces*—0.5%) are typically isolated from soil, but some also from the human skin bacteriome. Gram-positive non-endospore-forming rods were rare (1.5%), and some genera connected to the normal skin bacteriome were isolated (*Brevibacterium*, *Corynebacterium*, *Lactobacillus*—all 0.5%).

The bacterial load (CFU isolated from a sample) was higher than the fungal load. As 25% of samples were negative for bacterial contamination (no CFU isolated), in 47% of samples a maximum of 10 CFU were isolated. More than 10 CFU were isolated from 28% samples (11–30 CFU: 15%; >30 CFU: 13%, maximum 103 CFU). 

To determine the effect of different factors, the same approach as used for the fungal contamination was applied. Being higher, the total bacterial load was divided into four categories as follows: 0 CFU as no contamination; 1–10 CFU as low contamination; 11–30 CFU as moderate contamination; and more than 30 CFU as high contamination.

From one sample, a maximum of six genera or species (including unidentified isolates) were isolated. One genus was isolated in 38% of all samples (35% for one species), two genera in 17% (13% for two species), three genera in 13% (15% for three species), four genera in 5% (10% for four species), five genera in 1% (2% for five species) and six genera in 1% (3% for six species).

There was a statistically significant association (Pearson’s Chi-squared test, *p* = 0.05) between the number of bacterial CFU and the number of bacterial genera or species (*p* < 0.0001). The low level of bacterial contamination (1–10 CFU) was positively associated with the presence of one to three genera (except for one outlier sample with six genera). A moderate level of contamination (11–30 CFU) was positively associated with one to four genera or two to four species. High contamination (more than 30 CFU) was positively associated with one to three present genera or one to four present species, as there were some single species samples with extremely abundant growth.

Regarding the bacterial load in different kinds of samples (see Figure 1), PHPOS was more likely to be contaminated (77% of samples) than CIN and PHNEG (71% and 68%, respectively). However, if contaminated, a moderate or high level was more likely for CIN and PHPOS (30% and 29%, respectively, of samples) than for PHNEG (22%). 

No statistically significant association (Pearson’s Chi-squared test, *p* = 0.05) was found between the total bacterial load and different kinds of materials (*p* = 0.98), the material of the light-sensitive layer (*p* = 0.68) or the carrier (*p* = 0.59), but there was a significant association with archives from which samples were isolated (*p* = 0.004).

For bacterial genera, the effects of these factors were again analysed only for genera isolated in more than one sample (equal to more than one unique isolate), which included the genera *Staphylococcus* (39 isolates), *Bacillus* (25), *Streptococcus* (22), *Micrococcus* (*M*. *luteus*; 19 isolates), *Kocuria* (11), *Dermacoccus* (6), *Paenibacillus* (5), *Rothia* (5), *Neisseria* (5) and *Arthrobacter* (3).

No statistically significant association (Pearson’s Chi-squared test, *p* = 0.05) was found between these frequent genera and different kinds of materials (*p* = 0.30), the material of the light-sensitive layer (*p* = 0.60) and carrier (*p* = 0.22), or the combination of all these (*p* = 0.65). However, there was a statistically significant association between the species and the archives (*p* = 0.08) or the CFU isolated from a sample (*p* = 0.0008), and at level *p* = 0.10, also between these genera and archives (*p* = 0.08). However, the standardized Pearson residuals showed some strong positive and negative associations, as follows (see Figure 3); the corresponding relative frequencies are demonstrated in Appendix A.

Bold means XXX the frequency of the contamination for single genera isolated in more than 10% archives (for the genus *Micrococcus* is equal to the frequency of the species *M. luteus*).

The most often isolated genus *Staphylococcus* (90% archives) was ubiquitous, being isolated in all light-sensitive layer and carrier materials and their combination (15–100%)—except for GEL/CN (PHNEG; 0%). The genus *Streptococcus* was isolated less often (11–33%) and was not isolated from the PES carrier (GEL/PES, PHNEG and CIN) or the combination of COL/PAP (PHPOS) or GEL/CA (PHNEG). 

The genus *Bacillus* isolation (90% of archives) was positively associated with GEL/BP, GEL/PAP and ALB/PAP (PHPOS; 29–33% of samples). For CIN and PHNEG, it was isolated only from GEL/CA. A similar profile was shown for *Micrococcus luteus* (70% of archives) and the genus *Kocuria* (50% of archives)—both contaminated PHPOS with PAP or BP carriers, where they were positively associated with COL/BP and GEL/BP and with COL/PAP and GEL/BP, respectively. Both were also positively associated with GEL/CA (PHNEG) and were rarely isolated from GEL/CA (CIN).

The genus *Dermacoccus* (40% archives) was strongly positively associated with GEL/BP (PHPOS) and GEL/GL (PHNEG) and the genus *Rothia* (40% archives) with ALB/PAP (PHPOS) or GEL/GL (PHNEG). The genus *Paenibacillus* (40% archives) was positively associated with GEL/BP and COL/PAP (PHPOS), the genus *Neisseria* (30% archives) with GEL/CN and GEL/GL (PHNEG), and the genus *Arthrobacter* (20% archives) with OTH/PAP (PHPOS) and GEL/CA and GEL/GL (PHNEG).

Rarely isolated genera (isolated only in one sample in one archive) were isolated from more diverse combinations of the light-sensitive layer and carrier materials, but mainly from PHPOS and GEL. From PHPOS, they isolated the genera *Acinetobacter*, *Actinomyces*, *Brevundimonas*, *Gordonia* and *Sphingomonas* (GEL/BP), *Brevibacillus*, *Corynebacterium* and *Sporosarcina* (GEL/PAP), *Streptomyces* (COL/PAP), *Leuconostoc* (ALB/PAP) and *Oerskovia* (OTH/PAP). From PHNEG, the genera *Psychrobacillus* (GEL/PES) and *Kytococcus* (GEL/GL) were isolated, while from CIN, only the genus *Lactobacillus* (GEL/CA) was isolated. Almost all of these were isolated as 1 CFU, except for *Psychrobacillus psychrodurans* (2 CFU), *Brevibacillus borstelensis* (12 CFU) and *Leuconostoc lactis* (13 CFU).

Four genera (*Arthrobacter*, *Dermacoccus*, *Neisseria*, *Rothia*) were isolated only at a low level (1–10 CFU), while the genera *Micrococcus* and *Staphylococcus* were mainly isolated at a low level (95% and 91%, respectively). However, the genera *Kocuria*, *Streptococcus* and *Paenibacillus* also tended to be isolated at a moderate or high level. The genus *Bacillus* tended to be isolated similarly across the whole range of contamination levels.

### 3.4. Bacterial and Fungal Co-Contamination 

No statistically significant association (Pearson’s Chi-squared test, *p* = 0.05) was found between the level of fungal and bacterial co-contamination (*p* = 0.47) or the presence of single bacterial genera (*p* = 0.19). Generally, 11% of samples were negative for both fungal and bacterial contamination, while 14% of samples were positive only for fungal contamination and 29% only for bacterial contamination (see Figure 4). However, the ability to be isolated under an increased presence of fungi (more than 6 CFU of fungi to be isolated) was positively associated with bacterial genera such as *Staphylococcus*, *Kocuria*, *Streptococcus*, *Sporosarcina* or *Psychrobacillus* (data not shown).

## 4. Discussion

In this study we investigated the microbial diversity of contaminated photographic and cinematographic materials in 10 archival funds (archives) in the Czech Republic in 2019–2021. For fungi, we isolated and identified 120 isolates of 12 different genera, and for bacteria, 207 isolates of 25 genera and 56 species or species groups.

Generally, the level of contamination resulted from additional factors. These factors were, at minimum, the concentration of contaminating microorganisms in the environment (in the air, on human hands skin for manual handling), the sample exposure to this environment, the ability of microorganisms to survive on the surface of samples under unfavourable conditions and their ability to utilize a material and grow on it. All these effects were considered during analysis of the results, where single factors such as the archives (differences in environmental and storage conditions, handling archival items or disinfectant usage), type of material (different methods of storage or possible manual handling for CIN, PHPOS or PHNEG), type of material for the light-sensitive layer or the carrier (different chemical composition and degradability, antimicrobial properties, respective impact on changes in the light-sensitive layer, protection of light-sensitive layer from back side) were taken into account. Due to this, the samples were categorized, according to their type and the materials of the light-sensitive layer and carrier, into single groups.

The two most often isolated genera of fungi—the genera *Cladosporium* (40 samples) and *Penicillium* (18 samples)—were found to be isolated across almost all groups of samples. A similar situation could also be considered for the genera *Eurotium*, *Aspergillus* and *Alternaria*; however, for these genera, some specific facts were observed which require further investigation. As *Eurotium* was not isolated from collodion light-sensitive layers, it is under question whether collodion may not decrease its viability. On the other hand, the genera *Aspergillus* and *Alternaria* were isolated mainly from samples with paper or baryta paper (on lower level) as carriers. This raises the question of whether paper (or baryta paper) itself could not be another important source of contamination or a positive factor for fungi survival. Polyester as a carrier proved to be easily contaminated by different fungal genera. Gelatine layers were found to be an important factor for fungal diversity as they hosted all the rarely isolated fungal genera. 

Our results are comparable with other studies concerning the fungal contamination of cinematographic films in archives, such as studies from Opela for Slovak Republic archives [40] and from Abrusci et al., for Spanish archives in Madrid, Barcelona and Gran Canaria [28]. Abrusci et al. [28] isolated 17 strains of filamentous fungi, identified as the genera *Aspergillus* (four strains), *Penicillium* (seven strains) and *Alternaria*, *Cladosporium*, *Mucor*, *Phoma* and *Trichoderma*. In our study, we isolated CIN from most of them, but in different frequencies, such as the genera *Cladosporium* (six strains), *Eurotium* (three strains)/*Aspergillus* (one strain), *Alternaria* (one strain) and *Penicillium* (one strain). *Phoma* sp. was only isolated from one sample of PHPOS (gelatine on paper).

As the most often isolated fungal genera were isolated across most, or many of, the archives—such as the genera *Cladosporium* (90% of archives), *Penicillium* (60%), *Eurotium* (60%), *Aspergillus* (50%) and *Alternaria* (30%)—this raises the question of what the main source of this contamination is. A ubiquitous distribution may indicate that air is the main source for their contamination, being able to penetrate all materials, and the level of their contamination corresponding to their concentration in the air. In another study, Branysova et al. [41] analysed the fungal contamination of samples and air in four tested archives (C, D1, D2, E and F) using culture-independent methods (Illumina sequencing of DNA extracted from swab and air samples). The results obtained in our study using cultivation methods are comparable with the results obtained by Branysova et al. [41] using culture-independent methods, as the genera *Alternaria*, *Aspergillus* (in sequencing methods including the genus *Eurotium* as its teleomorph), *Cladosporium* and *Penicillium* were the most frequently detected in swab samples but were additionally also frequently detected in air samples.

The high frequency of the genus *Cladosporium* corresponds to study of Gorny and Dutkiewicz [42], who performed a general study on the biodiversity of aerosols in indoor environments in Central and Eastern European countries. In both these studies by Opela [40] and Gorny and Dutkiewicz [42], the genera *Aspergillus*, *Penicillium* and *Cladosporium* were the most frequently isolated genera. Abrusci et al. [28] explained their partially different results by the higher climatic differences that existed between the investigated Spanish archives (with a higher humidity in coastal areas in comparison to Madrid). 

Similarly, Vivar et al. [9], who analysed the fungal contamination of six samples of colour cinematographic films from Cuba archives ICAIC, isolated the genus *Aspergillus* more frequently than the other isolated genera *Cladosporium*, *Microascus* and *Penicillium*. Puskarova et al. isolated the fungal genera *Aspergillus* (species *A. versicolor*) and *Penicillium* from albumen photographic positives in the Slovak National Archives [16]. In contrast to Abrusci et al. [28], who also isolated *Cryptococcus albidus*, we did not isolate any yeast.

For bacterial contamination, Gram-positive bacteria were dominantly isolated—especially Gram-positive cocci such as the genera *Staphylococcus*, *Streptococcus*, *Kocuria*, *Micrococcus* and *Rothia,* or Gram-positive rods forming endospores such as the genus *Bacillus*. Gram-negative bacterial genera were isolated very sporadically, with the genus *Neisseria* being the most often isolated. 

These findings also correspond to Abrusci et al. [28], who identified 14 bacterial strains in Spanish archives from contaminated black and white cinematographic film samples. Five strains belonged to the genus *Staphylococcus* (*S. epidermidis*, *S. hominis*, *S. lentus*, *S. haemolyticus* and *S. lugdunensis*), five to the genus *Bacillus* (*B. amyloliquefaciens*, *B. subtilis*, *B. megaterium*, *B. pichinotyi* and *B. pumilus*) and one strain each to the species *Kocuria kristinae*, *Sphingomonas paucimobilis*, and *Pasteurella haemolytica*. Comparable to our study, Gram-positive bacteria of the genera *Staphylococcus* and *Bacillus* were dominant to Gram-negative bacteria. 

The high frequency of these Gram-positive bacteria genera corresponds to the study by Gorny and Dutkiewicz [42] on the biodiversity of aerosols in indoor environments in Central and Eastern European countries. In this study [42], Gram-positive cocci such as the genera *Micrococcus*, *Kocuria* and *Staphylococcus,* or Gram-positive rods forming endospores such as the genus *Bacillus* most frequently occurred. On the other hand, the genera *Pseudomonas* or *Aeromonas* were the most frequently occurring Gram-negative bacteria in this study [42] and were not isolated from archives samples.

Similarly, to the fungal genera *Cladosporium* and *Penicillium,* the most often isolated bacterial genera were also isolated across most archives, such as the genera *Bacillus* (90% of archives), *Staphylococcus* (90%), *Micrococcus* (70%) and *Kocuria* (50%). This might suggest the hypothesis that air should be also considered the main source for these bacterial genera contamination. As the genera *Staphylococcus*, *Micrococcus* and *Kocuria* are important parts of the normal human skin bacteriome, this may suggest that contamination via direct contact with human skin by manual handling may be possible but is not the dominant contamination vehicle. However, contrarily, the high occurrence of the genus *Streptococcus* (90% of archives), which is also an important part of the normal human skin bacteriome, does not correspond to its only occasional isolation from the air in indoor environments found by Gorny and Dutkiewicz [42]. To specify the exact impact of the air bacteriome in archives on the bacterial contamination of archival samples requires further investigation, as the diversity of the air bacteriome in archives may differ from that of other indoor environments (e.g., due to the controlled humidity and temperature in archives).

Similarly, to some fungal genera, the bacterial genera *Bacillus*, *Micrococcus* and *Kocuria* were isolated mainly from samples with paper or baryta paper as carriers. This also raises for these bacterial genera the question of what the precise impact of paper or baryta paper on the bacterial contamination of photographic and cinematographic materials is. 

Photographic and cinematographic negatives always had gelatine as the light-sensitive layer, but they differed in the material of the carrier. For bacteria, only the genus *Staphylococcus* was isolated from samples with a polyester carrier, and only the genera *Neisseria* and *Streptococcus* from samples with cellulose nitrate carriers. On the other hand, all frequent bacterial genera, except the genus *Rothia*, were isolated from samples with cellulose acetate carriers. For fungi, this carrier material effect in photographic and cinematographic negatives was more diverse. From samples with polyester carriers, all frequent fungal genera, except the genus *Alternaria*, were isolated—despite these samples being rare (a total of three samples from three different archives). While all frequent fungal genera were isolated from samples with cellulose acetate carriers, the genus *Cladosporium* was also frequently isolated from rare samples with cellulose nitrate acetate (three samples from three different archives). This fact should reflect the differences in the enzymatic activities of single bacterial and fungal genera, species, or even strains. Biochemical testing as a test for cellulase activity [16], the gelatine hydrolysis test [28] or DNA analysis of isolated strains would enable the evaluation of these differences and their impact on microbial contamination. 

Another interesting point for the further investigation of isolated microbial strains is to study their resistance to disinfectants or other antimicrobial compounds such as antibiotics, or to correlate their presence with the level of sample deterioration. Fungi are producers of different extracellular antimicrobial compounds (such as, e.g., *Penicillium* sp.). A possible resistance to antimicrobial compounds might explain the increased ability of some bacterial species (e.g., the genus *Staphylococcus*) for fungal co-contamination.

## Figures and Tables

**Figure 1 microorganisms-10-00155-f001:**
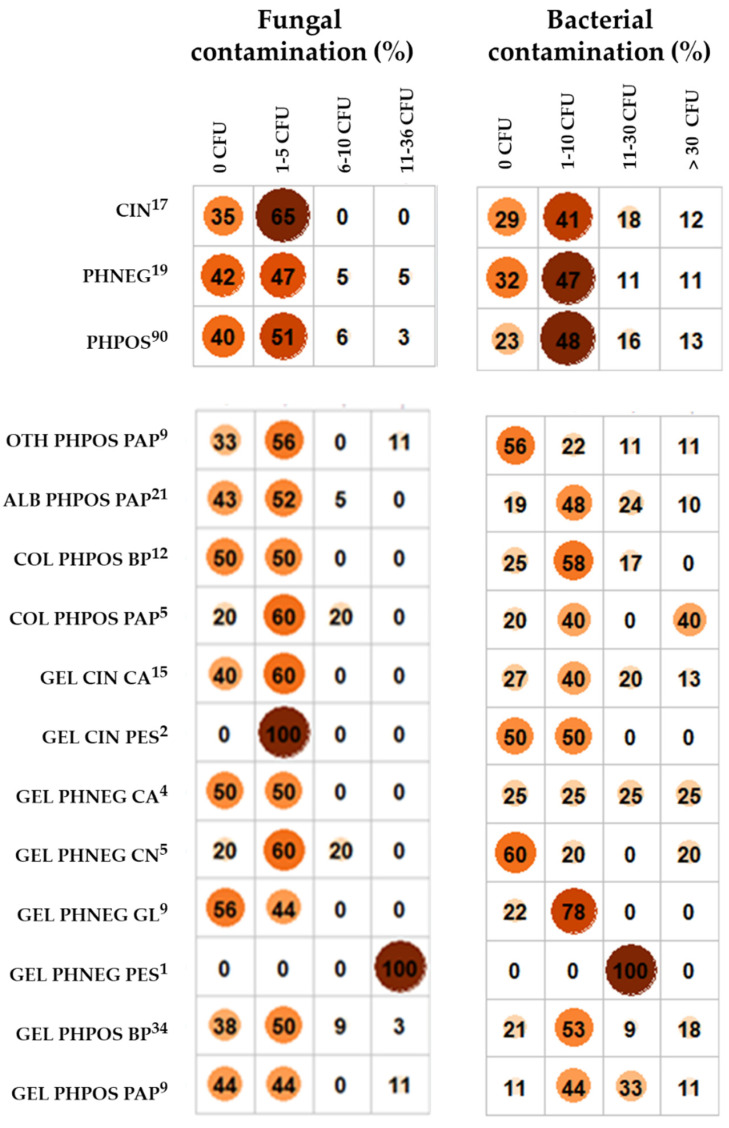
The relative frequencies of the categories of total fungal and total bacterial load (the number of fungal or bacterial CFU isolated from a sample) for samples categorised by the type of material (CIN, PHPOS, PHNEG) and by a combination of the type of material and the material of the light-sensitive layer and the carrier. Legend: CIN—cinematographic material, PHNEG—photographic negative, PHPOS—photographic positive, GEL—gelatine, ALB—albumen, COL—collodion, OTH—other minor types (such as autotype, ozotype, collotype, cyanotype, platinotype, salt paper and gumoil printing), CA—cellulose acetate, PES—polyester, CN—cellulose nitrate, PAP—paper, BP—baryta paper, and GL—glass. The upper index indicates the number of analysed samples.

**Figure 2 microorganisms-10-00155-f002:**
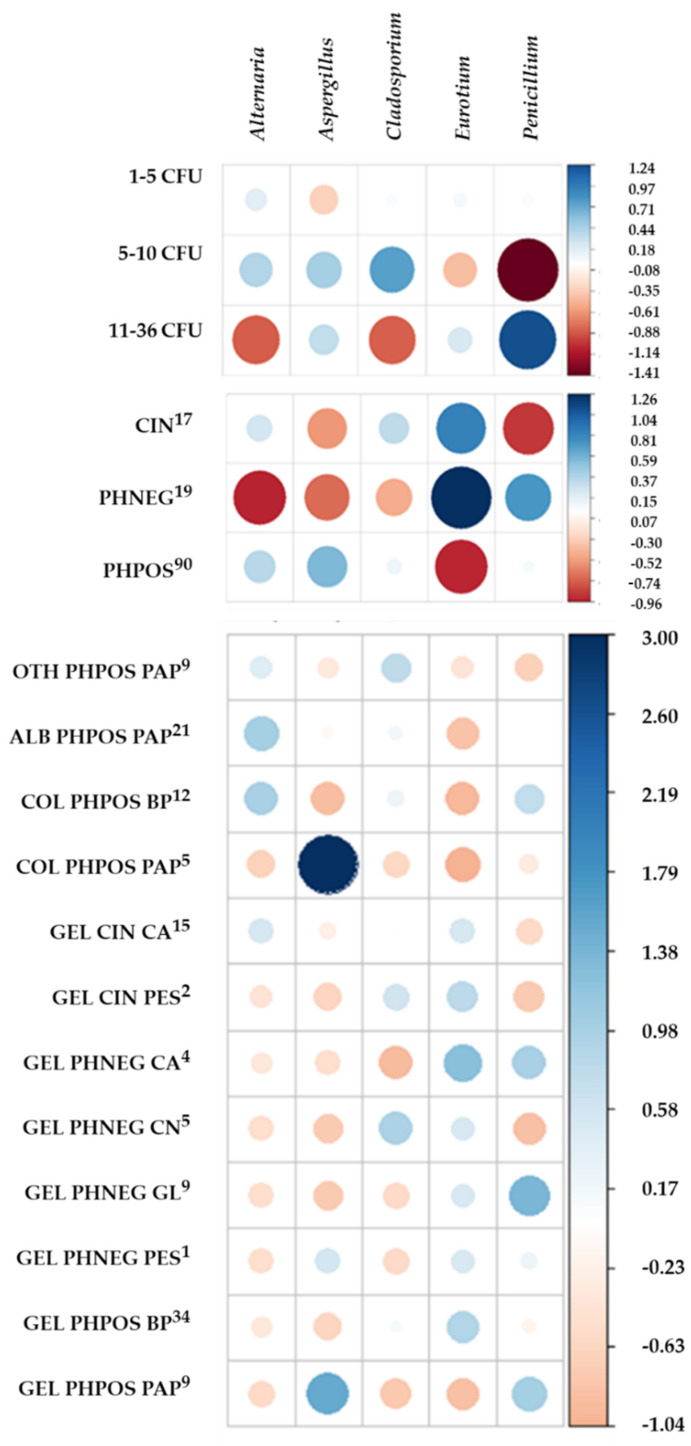
The standardized Pearson residuals for positive (blue) and negative (red) associations of the five most frequent fungal genera with analysed factors. These associations are between the fungal genus and the number of its CFU isolated from a sample, its isolation from single kinds of material (CIN, PHNEG, PHPOS) and the combination of the sample type and the materials of the light-sensitive layer and carrier. The size of the circle is proportional to the degree of cell contribution. The upper index indicates the number of analysed samples. Legend: CIN—cinematographic material, PHNEG-photographic negative, PHPOS—photographic positive, GEL—gelatine, ALB—albumen, COL—collodion, OTH—other minor kinds (such as autotype, ozotype, collotype, cyanotype, platinotype, salt paper and gumoil printing), CA—cellulose acetate, PES—polyester, CN—cellulose nitrate, PAP—paper, BP—baryta paper, GL—glass.

**Figure 3 microorganisms-10-00155-f003:**
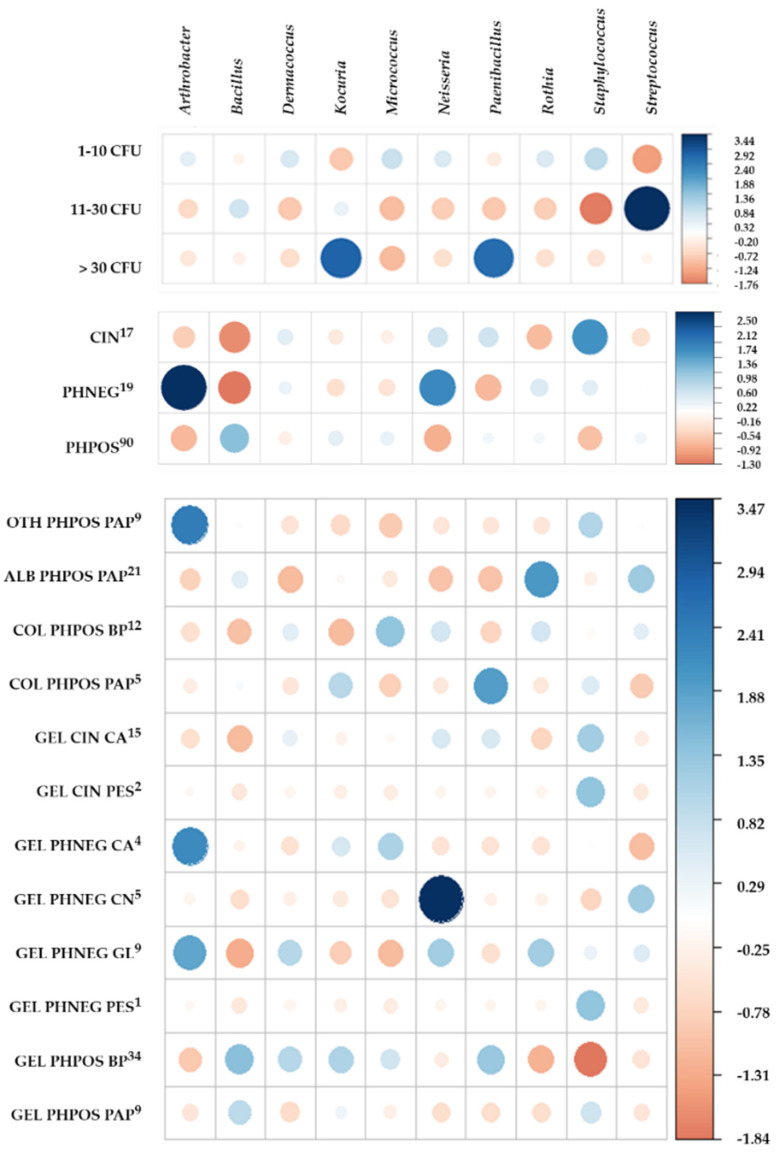
The standardized Pearson residuals for positive (blue) and negative (red) associations of the ten most frequent bacterial genera with analysed factors. These associations are between the bacterial genus and the number of its CFU isolated from a sample (regarding single species), its isolation from single kinds of material (CIN, PHNEG, PHPOS) and the combination of the sample type and the material of the light-sensitive layer and carrier. The size of the circle is proportional to the degree of cell contribution. The upper index indicates the number of analysed samples. Legend: CIN—cinematographic material, PHNEG—photographic negative, PHPOS—photographic positive, GEL—gelatine, ALB—albumen, COL—collodion, OTH—other minor kinds (such as autotype, ozotype, collotype, cyanotype, platinotype, salt paper and gumoil printing), CA—cellulose acetate, PES—polyester, CN—cellulose nitrate, PAP—paper, BP—baryta paper, GL—glass.

**Figure 4 microorganisms-10-00155-f004:**
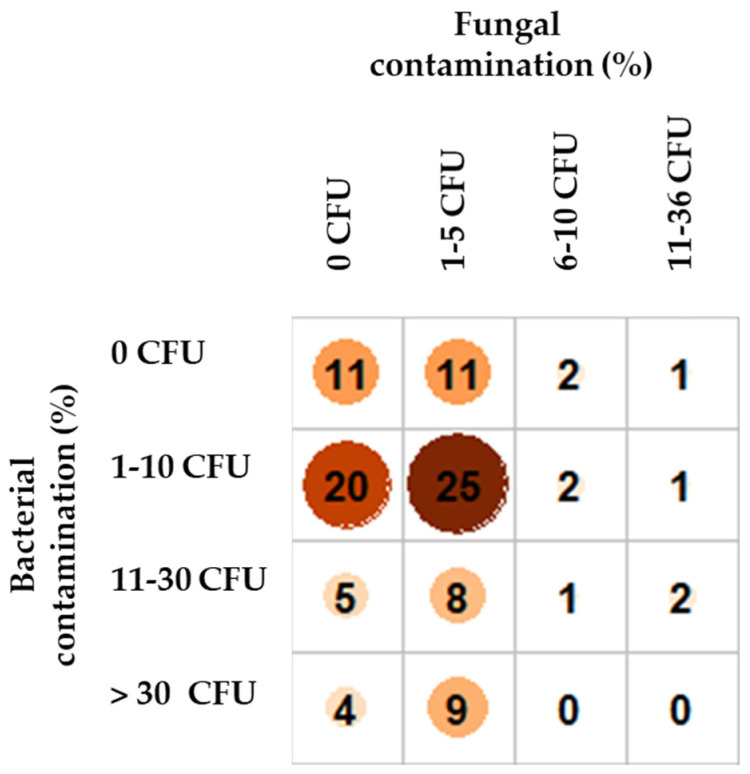
The frequency of single levels of bacterial and fungal contamination in tested samples.

**Table 1 microorganisms-10-00155-t001:** Fungal contamination from single archives regarding the percentage of samples positive for fungal contamination (CFU ≥ 1) and genera and species present.

Archives	G2	C	D2	G3	G1	A	E	F	D1	B	Sum
Samples (no.)	7	22	8	13	3	17	15	10	13	18	126
Identified genera (no.)	1	8	2	5	1	4	4	3	6	2	12
Positive samples (%)	100	91	88	85	67	47	47	40	38	28	60
*Cladosporium* sp.	100 ^6^	32	38 ^6^	77	67	18	27	30	8	.	90
*Eurotium* sp.	.	23	.	8 ^7^	.	12	20	.	8	11	60
*Penicillium* sp.	.	18	.	8	.	18	27	20	31 ^8^	.	60
*Aspergillus* sp.	.	.	75 ^2,3^	.	.	12 ^1^	13 ^4^	.	15 ^2^	6 ^1^	50
*Alternaria* sp.	.	18 ^5^	.	8 ^5^	.	.	.	10	.	.	30
*Arthrinium* sp.	.	5	.	.	.	.	.	.	.	.	10
*Epicoccum nigrum*	.	5	.	.	.	.	.	.	.	.	10
*Geomyces* sp.	.	5	.	.	.	.	.	.	.	.	10
*Tritirachium oryzae*	.	5	.	.	.	.	.	.	.	.	10
*Aureobasidium pullulans*	.	.	.	8	.	.	.	.	.	.	10
*Phoma* sp.	.	.	.	.	.	.	.	.	8	.	10
*Wallemia sebi*	.	.	.	.	.	.	.	.	8	.	10
NID ^9^	29	36	25	15	.	6	20	.	8	11	80

^1^—*Aspergillus* section Nigri (B: 6%), ^2^—*Aspergillus versicolor* (D1: 7.5%, D2: 50%), ^3^—*Aspergillus* spp. (different species from one sample; D2: 12.5%), ^4^—*Aspergillus niger* (D2: 7.5%), ^5^—*Alternaria* spp. (different species from one sample; C: 4.5%, G3: 8%), ^6^—*Cladosporium* spp. (different species from one sample; D2: 25%, G2: 28.5%, G3: 31%), ^7^—*Eurotium* spp. (different species from one sample; G3: 8%), ^8^—*Eurotium* spp. (different species from one sample; D1: 31%), ^9^—number of unidentified isolates (archives—number of all unidentified isolates): A—1, B—2, C—8, D1—1, D2—2, E—3, F—0, G1—0, G2—2, G3—2.

**Table 2 microorganisms-10-00155-t002:** The frequency of bacterial contamination (for present genera and species) from single archives regarding the percentages of samples positive for bacterial contamination (CFU ≥ 1).

Archives	A	G1	F	B	G3	D2	E	D1	C	G2	Sum
Samples	17	3	10	18	13	8	15	13	22	7	126
Identified genera (no.)	7	3	10	12	6	6	8	10	5	4	25
Identified species (no.)	17	8	16	20	12	14	14	15	12	6	56
Positive samples (%)	100	100	80	78	77	75	67	62	59	43	75
** *Staphylococcus* ** **sp.**	**47**	**33**	**10**	**22**	**15**	**50**	**53**	**38**	**27**	**.**	**90**
*S. epidermidis*	12	33	10	11	.	13	20	8	5	.	80
*S. hominis*	24	.	10	6	.	.	27	.	14	.	50
*S. warneri*	6	.	.	6	.	25	.	8	9	.	50
*S. capitis*	6	.	.	.	.	13	13	23	.	.	40
*S. aureus*	6	.	.	.	15	.	.	.	.	.	20
*S. caprae*	.	.	.	.	.	.	7	.	.	.	10
*S. haemolyticus*	.	.	.	.	.	.	7	.	.	.	10
*S. pasteuri*	.	.	.	.	.	13	.	.	.	.	10
** *Streptococcus* ** **sp.**	**18**	**.**	**10**	**22**	**46**	**25**	**7**	**8**	**9**	**29**	**90**
*S. salivarius*	18	.	10	6	46	13	.	8	5	29	80
*S. parasanguinis*	6	.	10	17	8	13	.	.	5	14	70
*S. pneumoniae group* ^1^	.	.	.	.	8	25	7	.	5	14	50
*S. vestibularis*	6	.	.	11	8	.	.	.	.	.	30
*S. sanguinis*	.	.	.	.	.	.	.	.	5	.	10
** *Bacillus* ** **sp.**	**29**	**100**	**20**	**22**	**31**	**25**	**13**	**15**	**.**	**14**	**90**
*B.* cereus group ^2^	6	33	.	11	15	25	7	.	.	.	60
*B. pumilus*	6	.	10	.	8	.	7	8	.	.	50
*B. licheniformis*	12	.	.	.	.	.	.	8	.	14	30
*B. megaterium*	.	67	10	.	.	25	.	.	.	.	30
*B. muralis*	.	33	.	6	.	.	7	.	.	.	30
*B. subtilis*	12	.	10	.	15	.	.	.	.	.	30
*B. sonorensis*	.	.	10	.	.	.	.	8	.	.	20
*B. atrophaeus*	.	33	.	.	.	.	.	.	.	.	10
*B. circulans*	.	33	.	.	.	.	.	.	.	.	10
*B. clausii*	.	.	10	.	.	.	.	.	.	.	10
*B. simplex*	.	.	.	6	.	.	.	.	.	.	10
*B. thermoamylovorans*	.	.	.	.	.	13	.	.	.	.	10
** *Micrococcus luteus* **	**35**	**.**	**10**	**39**	**.**	**13**	**13**	**.**	**5**	**14**	**70**
** *Kocuria* ** **sp.**	**.**	**.**	**.**	**6**	**23**	**13**	**.**	**15**	**18**	**.**	**50**
*K. palustris*	.	.	.	.	.	.	.	.	5	.	10
*K. rhizophila*	.	.	.	6	23	13	.	15	14	.	50
** *Dermacoccus* ** **sp.**	**12**	**.**	**10**	**11**	**.**	**.**	**.**	**.**	**5**	**.**	**40**
*D. nishinomiyaensis*	12	.	.	11	.	.	.	.	5	.	30
*D. profundi*	.	.	10	.	.	.	.	.	.	.	10
** *Paenibacillus* ** **sp.**	**6**	**33**	**.**	**11**	**.**	**.**	**.**	**8**	**.**	**.**	**40**
*P. glucanolyticus*	6	.	.	.	.	.	.	8	.	.	20
*P. amylolyticus*	.	33	.	.	.	.	.	.	.	.	10
*P. lactis*	.	.	.	6	.	.	.	.	.	.	10
*P. macerans*	.	.	.	6	.	.	.	.	.	.	10
*P. residui*	.	33	.	.	.	.	.	.	.	.	10
** *Rothia* ** **sp.**	**.**	**.**	**10**	**.**	**15**	**.**	**.**	**8**	**.**	**14**	**40**
*R. dentocariosa*	.	.	.	.	15	.	.	8	.	14	30
*R. amarae*	.	.	10	.	.	.	.	.	.	.	10
** *Neisseria* ** **sp.**	**12**	**.**	**10**	**11**	**.**	**.**	**.**	**.**	**.**	**.**	**30**
*N. flavescens*	6	.	10	11	.	.	.	.	.	.	30
*N. perflava*	6	.	.	.	.	.	.	.	.	.	10
** *Arthrobacter* ** **sp.**	**.**	**.**	**.**	**6**	**.**	**.**	**13**	**.**	**.**	**.**	**20**
*A. luteolus*	.	.	.	.	.	.	13	.	.	.	10
*A. polychromogenes*	.	.	.	6	.	.	.	.	.	.	10
*Brevibacterium luteolum*	.	.	10	.	.	.	.	.	.	.	10
*Corynebacterium* *lipophiloflavum*	.	.	10	.	.	.	.	.	.	.	10
*Kytococcus sedentarius*	.	.	10	.	.	.	.	.	.	.	10
*Acinetobacter lwoffii*	.	.	.	6	.	.	.	.	.	.	10
*Actinomyces oris*	.	.	.	6	.	.	.	.	.	.	10
*Gordonia aichiensis*	.	.	.	6	.	.	.	.	.	.	10
*Brevundimonas aurantiaca*	.	.	.	.	8	.	.	.	.	.	10
*Brevibacillus borstelensis*	.	.	.	.	.	13	.	.	.	.	10
*Oerskovia turbata*	.	.	.	.	.	.	7	.	.	.	10
*Psychrobacillus psychrodurans*	.	.	.	.	.	.	7	.	.	.	10
*Sphingomonas* *pseudosanguinis*	.	.	.	.	.	.	7	.	.	.	10
*Lactobacillus oligofermentans*	.	.	.	.	.	.	.	8	.	.	10
*Leuconostoc lactis*	.	.	.	.	.	.	.	8	.	.	10
*Sporosarcina luteola*	.	.	.	.	.	.	.	8	.	.	10
*Streptomyces hirsutus*	.	.	.	.	.	.	.	8	.	.	10
NID ^3^	.	67	0	17	38	50	7	8	9	.	70

^1^–isolates of species *Streptococcus mitis*/*oralis*/*peroris*/*pneumoniae*, ^2^—isolates of species *Bacillus cereus*/*thuringiensis*/*weihenstephanensis*, ^3^—number of unidentified isolates (archives—number of all unidentified isolates): A—0, B—3, C—2, D1—1, D2—4, E—1, F—0, G1—2, G2—0, G3—5.

## Data Availability

Data are available upon request to the corresponding authors.

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
