# Peer review of "Microbial Contamination of Photographic and Cinematographic Materials in Archival Funds in the Czech Republic"

_microorganisms, 2022, doi:10.3390/microorganisms10010155_

Round 1

Reviewer 1 Report

The objective of this study “To investigate microbial contamination of cinematographic and photographic archive materials” seems to me very original and exciting. The authors analyzed 126 samples and I found their efforts and obtained results very valuable, however there are major issues to be considered and to be fixed before publication of this paper.

Introduction:

The whole introduction must be reformulated, because to me it doesn’t fit to the main purpose of the paper. For example, the major part of the introduction discusses the photography techniques and the technology, or how carefully to handle and preserve the archive material.  To be honest, I have enjoyed reading all these details, but I would rather prefer to get background about discussed microorganisms, similar studies in the field related to these microorganisms, the objectives and the needs for this study design, why this study design is more innovative and what outcome they expect, etc.  

Material and methods:

Write full culture  media names like  MEA, DG18 etc.

 I would organize the microorganism’s identification parts in 2 subsection describing how you identify fungi and bacteria separately

Results:

I am not sure if the first part of 3.1 is relevant to the objective of the paper.  

In 3.2.

83 % (99 isolates) were identified at least on the genus level (12 genera) and 17 % (21 isolates) were not identified as they did not sporulate. => wouldn’t it be much accurate to perform molecular barcoding and accurately identify all genera and species?

The relative frequencies of samples positive for fungal contamination and present fungal genera in single archives are summarised in Error! Reference source not found.. => what does this  mean?

Some statements here  are not convincing, example

It should indicate that air is the main source for their contamination as to be able to penetrate to all materials and the level of their contamination should correspond to their concentration in the air. => Do you have any control culture, to detect what is in the air alone?  

Gelatin layer was found to be the important factor for fungal diversity as it hosted all rarely isolated fungal genera.=> How do you support this statement?

Discussion

The discussion is very poor, the second part of discussion is fitting more to introduction.

There is no actual hypothesis, mainly it is saying that their results are similar to the outcomes from other studies. Does this study found something new, does your design and methods add anything new to the already existing data?

It would be great to see actual comparative discussion/hypothesis  regarding type of archive, environmental conditions and microbial fauna. I would also present it in the form of table. Finally if you know the contamination and the source of contamination, some brief suggestions how to prevent it would be valuable.

Reference list.

The reference list must be unified, there are many formatting/ style errors.

Reviewer 2 Report

A well presented, rigorous and descriptive work of the habitual microbiota on this type of cultural assets. In my opinion it can be published in the current format.
A continuation of it would be very interesting, dealing with the important question of assigning typology of damage to the various microorganisms described, and not limiting the question to the mere description of the microorganisms present at a specific moment.

Author Response

I really thank you for your high appreciation of our study and the recommendations for the further investigation.

Reviewer 3 Report

The paper by Purtova et al. is interesting and focuses on the biodeterioration of photographic and cinematographic materials.

The Introduction is well written but for me is a little too long. The aim of the study is well pointed out.

Materials and Methods are more or less explained but for me the number of culture media the authors used for isolation is rather low. Only a small fraction of the culturable microorganisms (fungi and bacteria) would show up with this number of culture media. Also, the number of days of isolation for me is too short, considering some groups of fungi (that can be slow growers). I would have increased the period of isolation.

Results are, I think, limited by the methodology the authors employed. As for the statistical analysis, it seems correct to me, but I wonder if the authors tried to employ PCA analysis to see if the same significant associations or other would come out.

How the authors explain as for bacteria the loads are much higher than for fungi? Can this be a reflex of the sampling and isolation procedures?

Discussion is extensive and well-articulated. However, it would be important to me if the authors could correlate the level of deterioration in some supports and items with the presence of some species (bacteria or fungi or both groups). The authors point out this in the last paragraph of the discussion. Also as the authors say, it would be important to perform biochemical testing as for example test for cellulase activity or gelatin hydrolysis to ascertain the biodeteriorative potential of some of the isolated microorganism in each type of support.

The number of references (only 26) is rather low in my opinion!

I have also several concerns and questions:

1.How was planned the sampling? What was the rationale for this sampling, I mean the number of collections, the number of samples per collection? Was it random? I acknowledge that samples/items suspected for microbial contamination were chosen and this I think was the main criterion.

  1. The authors do not describe in a complete and proper manner how they identified the fungi. They mention that fungi were identified by microscopy and appropriate biochemical and physiological testing (no references are given!). This is very vague to me. Only by observation of morphological structures is impossible to reach the correct identification of the species in many fungal groups! Why did not the authors apply modern molecular techniques? Moreover, grouping the isolates only based on morphological features and aspect of the colony is sometimes insufficient…This can be the reason for the low diversity found (few genera found) and for the low number of isolates concerning the all samples (see 3.2). Also in this section there is an “Error stated in the last phrase”! I think it is missing the last part of the phrase, indicating Table 1…

Also, in some genera it is crucial to determine the species as this can be important for public health reasons.

On the other hand, for bacteria the authors employed MALDI-TOF MS identification and this seems adequate or at least satisfactory to me!

Nevertheless, it is important to highlight the point about Bacterial and fungal co-contamination. This is a very important result for me!

  1. The authors refer that air can be a vehicle of contamination (see abstract and discussion). Were air samples taken and analysed? I believe not (nothing is said about this, so this is only an assumption, not a fact! Please be careful about this!

See this statement of yours: “This bacterial and fungal diversity refers to air as the main vehicle of contamination.”

  1. Why did not the authors perform metagenomics analyses of the microbiome that was present at least in some samples? They compare some studies where these NGS analyses were employed. I do think that both types of approaches using culturable dependent and independent methods are important and they should be used together.

Round 2

Reviewer 1 Report

I appreciate the authors efforts to answer my questions and to make the corrections. Authors did not carry out further analysis to fulfill two of my suggestions (Point 6 and Point 11), which in my opinion would made this study stronger and interesting for broader audience. Nevertheless, I understand and accept their arguments about their limited possibilities and wish to conduct a separate study focusing more on the separate environmental factors. As such, I think it still useful to get insights of microbial diversity from these particular archive material and I recommend this manuscript for the publication.  

Reviewer 3 Report

For me, the paper has been improved in the possible way. All my major concerns were properly answered and explained. Nevertheless, the Introduction for me remains quite long. Moreover, the authors added three new paragraphs (lines 212-251). Please check english and grammar in these 3 new paragraphs (especially lines 225-230). Also, in future rebuttals please check english and be more careful regarding the language.